# New Perspectives on Primary Prophylaxis of Invasive Fungal Infection in Children Undergoing Hematopoietic Stem Cell Transplantation: A 10-Year Retrospective Cohort Study

**DOI:** 10.3390/cancers15072107

**Published:** 2023-03-31

**Authors:** Noémi Ricard, Lelia Zebali, Cécile Renard, Marie-Pierre Goutagny, Sarah Benezech, Yves Bertrand, Michael Philippe, Carine Domenech

**Affiliations:** 1Centre Léon Bérard, Pharmacy Department, 69008 Lyon, France; 2The Pediatric Hematology and Oncology Institute, Hospices Civils de Lyon, Université Lyon 1, 69008 Lyon, France; lelia.zebali@gmail.com (L.Z.);; 3Faculty of Medicine Lyon Est, Université Claude Bernard Lyon 1, 69008 Lyon, France; 4Faculty of Medicine and Maieutics Charles Mérieux Lyon Sud, Université Claude Bernard Lyon 1, 69921 Lyon, France; 5International Center of Research in Infectiology, Université Lyon 1, INSERM U 1111, CNRS UMR 5308, 69007 Lyon, France

**Keywords:** invasive fungal infection, hematological diseases, hematopoietic stem cell transplantation, children, antifungal prophylaxis

## Abstract

**Simple Summary:**

In children undergoing allogenic HCT, invasive fungal infections are a significant cause of mortality, for which ECIL-8 have proposed systematic primary prophylaxis. This study evaluates our local strategy of not systematizing it, and reveals a similar IFI incidence and mortality rate.

**Abstract:**

Background: Allogenic hematopoietic stem cell transplantation (a-HCT) remains a therapeutic treatment for many pediatric hematological diseases. The occurrence of invasive fungal infections (IFIs) is a complication for which ECIL-8 recommends primary antifungal prophylaxis. In this study, we evaluated the impact of our local strategy of not systematically administering primary antifungal prophylaxis in children undergoing a-HCT on the occurrence and mortality of IFIs. Methods: We performed a retrospective monocentric study from 2010 to 2020. We retained all proven and probable IFIs diagnosed during the first year post a-HCT. Results: 308 patients were included. Eighteen patients developed twenty IFIs (thirteen proven, seven probable) (6.5%) among which aspergillosis (*n* = 10, 50%) and candidosis (*n* = 7, 35%) were the most frequently diagnosed infections. Only 2% of children died because of an IFI, which represents 14% of all deaths. Multivariate analysis found that age > 10 years (OR: 0.29), the use of a therapeutic antiviral treatment (OR: 2.71) and a low neutrophil count reconstitution (OR: 0.93) were significantly associated with the risk of IFI occurrence. There was also a trend of malignant underlying disease and status ≥ CR2 but it was not retained in multivariate analysis. Conclusions: IFI occurrence was not higher in our cohort than what is reported in the literature with the use of systematic antifungal prophylaxis, with a good survival rate nonetheless. Thus, a prophylaxis could be considered for children with a high risk of IFI such as those aged over 10 years.

## 1. Introduction

Allogeneic hematopoietic stem cell transplantation (a-HCT) remains an effective treatment procedure for many children with hematological disorders. The French Agency of Biomedicine estimated that approximately 270 pediatric HCTs are annually performed in France [1]. Different pathologies could be concerned: malignant diseases such as leukemia or benign diseases such as sickle cell anemia, primary immunodeficiencies and metabolic syndromes [2].

Despite major advances in HCT procedures, the main complications remain as graft versus host disease (GvHD) and infection occurrence [3]. Invasive fungal infections (IFIs) have emerged as important causes of morbidity and mortality. Indeed, the incidence rate reported for IFIs varies from 1.2% to 25% depending on the study, with a mortality rate that can reach 80% in the most severe forms [4,5,6,7,8,9,10,11,12,13,14,15]. In these patients, infections with *Candida* and *Aspergillus* species are the most common, even if mucormycosis have emerged as increasingly relevant and highly lethal causes of IFIs in many centers worldwide [16,17,18]. The diagnosis of IFIs is difficult for several reasons: various but non-specific clinical symptoms, delicate isolation of fungal organisms, non-specific imaging and biological markers [19,20,21,22,23,24]. This often leads to a delay in therapeutic support and therefore poses a challenge to children’s survival.

According to ECIL-8 recommendations, antifungal prophylaxis is indicated in patient populations with a natural incidence rate ≥ 10%, including a-HCT recipients. Fluconazole is not recommended in a-HCT post-engraftment due to the predominant role of mold infections in this setting. ECIL-8 recommends the use of posaconazole (oral suspension for <13 years old children and adult tablets for >13 years old children) with level A evidence. Then, itraconazole, voriconazole and liposomal amphotericin b are recommended with level B evidence and next, micafungin with level C evidence [22]. However, as stated in the guideline, the local epidemiology is an important consideration to design local approaches for the management of invasive fungal diseases (IFDs), and considering patients’ individual risk factors is also essential for best management. The incidence of IFIs, specific local ecology, different pharmacokinetics and drug interactions in the pediatric population should also lead to an assessment of the benefit–risk balance before initiating this treatment on a routine basis [13,25]. In our tertiary reference center, the strategy is based on environmental prophylaxis during all the HCT procedures, without systematic primary prophylaxis. This environmental prophylaxis includes strict hygiene measures with hand disinfection to protect patients from hand-borne organisms and a protective laminar flow environment with HEPA (High Efficiency Particulate Air) filtration [26,27]. The objective of our monocentric retrospective study was to evaluate IFI incidence in a large cohort of children undergoing a-HCT from 2010 to 2020.

## 2. Materials and Methods

### 2.1. Study Population

This is a retrospective cohort study conducted in a tertiary reference center for hematological diseases welcoming children and young adults. All patients under 20 years of age who received an a-HCT in our center between 16 April 2010 and 19 March 2020 were included in this study. Data were obtained from the prospectively collected database ProMISe (The European Bone Marrow Transplantation Registry) and included the patient’s demographics, underlying diseases, HCT characteristics, immune and hematological recovery time, and diagnosis of IFIs. All informed consents were obtained before the a-HCT procedure, according to the reference methodology MR-004 R201-004-187.

### 2.2. Transplant Methods

Allogenic HCTs were performed according to the French Society of Bone Marrow Transplantation (SFGM-TC) guidelines. All patients were confined and cared for in high-efficiency, particle-free, air-filtered, positive-pressure isolation bedrooms from the beginning of the conditioning to the neutropenia recovery and for all prolonged neutropenia periods. A majority of patients received myeloablative conditioning (MAC) regimens with either total body irradiation (TBI) or busulfan. Patients received gut decontamination which consisted of mouth care (4 times a day) with the addition of vancomycin mouthwashes (2 times a day) to eliminate *Streptococcus* spp. [28]. They also benefited from a non-absorbable digestive decontamination with amoniglycoside (gentamycine or amikacin), or colimycin in presence of *Pseudomonas* spp. and amphotericin B (Fungizone^®^), only during the neutropenia period. Patients also received veno-occlusive disease prophylaxis (ursodesoxycholic acid), GvHD prophylaxis (cyclosporine alone with close pharmacokinetic monitoring [29], or, for a few patients, associated with mycophenolate mofetil) and infection prophylaxis (acyclovir, phenoxymethylpenicillin and trimethoprim–sulfamethoxazole). Only patients with previous IFI history systematically received secondary prophylaxis.

### 2.3. IFI Definition, Monitoring and Antifungal Treatment

The primary outcome analyzed was IFI incidence. IFIs were diagnosed according to Donnelly et al.’s classification. An IFI was proven if the presence of molds or yeasts as shown by histopathological, cytopathological or direct examination of a sample was obtained by aspiration or biopsy; by positive blood culture; by culture of a sample recovered by sterile procedure from a clinically or radiologically abnormal site (such as bronchoalveolar fluid); or by polymerase chain reaction (PCR) of sterile tissue. An IFI was considered probable if clinical and radiological criteria such as excavation or halo sign in a patient at risk, meaning immunosuppressed, were associated with mycological criteria such as antigenemia (galactomannan (GM) antigen) [30].

Patients with fever >38 °C received empiric antibacterial therapy. Investigations for persistent fever for 48–72 h, after the latest modification of antibacterial therapy, consisted of multiple fungal blood cultures, repeated serum galactomannan (GM) antigen and blood mucorales PCR, associated with a chest-X-ray and/or CT-scan in case of severe neutropenia. Bronchoalveolar lavage (BAL) was performed if pulmonary IFI was suspected. In all cases of suspected IFI, a direct examination according to the site of the infection (pleural punction, biopsy, sinus swab…) was performed whenever possible. Then, an empirical antifungal therapy was started and consisted preferentially in an echinocandin which is less toxic for renal function and causes less drug interactions with immunosuppressive agents. The empirical antifungal treatment could sometimes be modified, at the discretion of the attending physician. It was also the case for the duration of targeted antifungal treatment for IFI, which was continued at least until clinical symptom resolution, blood culture sterilization, and any measurable disease by high-resolution CT scan.

### 2.4. Clinical Outcomes and Definitions

Neutrophil engraftment was defined as an absolute neutrophil count greater than 500 cells/μL for 2 consecutive days; lymphocyte reconstitution when the cell count was greater than 1500 cells/μL for 2 consecutive samples.

High dose steroid therapy was defined as patients who received prednisolone with a minimal dosage of 1 mg/kg/day for more than a week.

Therapeutic antiviral treatment was defined as patients who received an antiviral drug such as rituximab, cidofovir, ganciclovir, ribavirine, brincidofovir or another, except acyclovir (used in infection prophylaxis).

GvHD was graded by standard clinical criteria and staged according to the Glucksberg et al. criteria. [31], modified by Thomas, E.D., et al. [32], including the extent of rash, daily diarrhea volume and serum bilirubin.

Secondary outcomes were the 1-year overall survival (OS), incidence of relapse and stay in an intensive care unit.

### 2.5. Descriptive Statistics and Logistic Regression Analysis

Frequencies, percentages and median value were used for descriptive statistics. The Chi-squared test was used to compare categorical variables and the Student *t*-test or the Mann–Whitney U-test was used for continuous variables, as appropriate. All tests were two-tailed with statistical significance set at 0.05.

Logistic regression analysis was performed to investigate the variables associated with IFIs occurrence. A stepwise procedure was carried out. All variables were first tested in univariate logistic regression. The statistical significance of the regression coefficient associated with each variable was assessed using the Wald test. Variables with a *p* value < 0.20 in the univariate analyses were included in the multivariate model. A backward stepwise procedure was then applied. We chose the final model according to the Akaike Information Criterion (AIC) which is a function of the log-likelihood penalized for over-parameterization. Adjusted odds ratios (OR) and their associated 95% confidence intervals (CIs) were calculated. All analyses were conducted using R software, version 4.1.2.

Overall survival was analyzed by the Kaplan–Meier method, and comparison between samples was performed using the log-rank test. *p* values < 0.05 were considered significant.

## 3. Results

### 3.1. Clinical Characteristics of Patients (Table 1)

A total of 308 children (67% male) were retrospectively included in this study. The median age (range) at the time of a-HCT was 8.5 years (one month to 20 years) and 61% of the population was younger than 10 years old. Most patients had a hematological malignancy (57%), with the majority having acute lymphoid leukemia (ALL). Almost 27% (83/308) of children had a complete remission (CR) status ≥2 before a-HCT. Most patients had an unrelated donor (61%) and had received a bone marrow transplant (74.3%), following a MAC regimen in the majority of cases (95.1%). The mean neutrophil recovery time was 22.9 days. Half of the patients (55.2%) received high dose steroid treatment after a-HCT with a mean duration of 90 days. One in five patients (20.5%) also received an additional second line immunosuppressive treatment; ruxolitinib and mycophenolate mofetil were the most immunosuppressive drugs used in our cohort. Lastly, almost 12% of patients relapsed and only 5% presented an engraftment failure.

**Table 1 cancers-15-02107-t001:** Clinical characteristics of 308 pediatric allogenic HCT recipients.

Patient Characterisitcs	Total No. of Patients (%)	Invasive Fungal Infection				*p*-Value	
		Yes			No		
		No. of Patients (%)			No. of Patients (%)		
	308		18 (5.8)		290 (94.2)		
Gender							
Male	184		8		176		
Female	124		10		114		
		sd		sd		sd	
Median patient Age [min-max]	8.6 (0.1–20)	5.3	11.9 (2–19)	5.6	8.4 (0.1–20)	5.2	0.0089
<10 years old	188 (61)		6 (33.3)		182 (62.4)		0.014
Indication for HCT							
Malignant disease	176 (57)		13 (72.2)		163 (56.2)		
ALL	86 (27.9)		5 (27.8)		81 (27.9)		
AML	50 (16.2)		2 (11.1)		48 (16.6)		
CR2 or more	83 (26.9)		7 (38.9)		76 (26.2)		0.022
Non-Malignant disease	127 (41.2)		5 (27.8)		122 (42.1)		
Primary immunodeficiencies	43 (14.0)		1 (5.5)		42 (14.5)		
Thalassemia	18 (5.8)		0 (0)		18 (6.2)		
Sickle cell disease	16 (5.2)		0 (0)		16 (5.5)		
Metabolic disorders	5 (1.6)		0 (0)		5 (1.7)		
Type of donor							
Matched related donor	106 (34.4)		6 (33.3)		100 (34.5)		
Unrelated donor	188 (61.0)		12 (66.6)		176 (60.7)		
Haplo-identical donor	14 (4.5)		0 (0)		14 (4.8)		
Stem cell source							
Bone marrow	229 (74.3)		11 (61.1)		218 (75.2)		
PBSC	19 (6.2)		2 (11.1)		17 (5.9)		
Cord blood unit	56 (18.2)		5 (27.8)		51 (17.6)		
Mixed	4 (1.3)		0 (0)		4 (1.4)		
Type of conditionning							
Myeloablative regimen	293 (95.1)		17 (94.4)		276 (95.2)		
With Busulfan	185 (60.1)		8 (44.4)		177 (61)		
With TBI	79 (25.6)		9 (50.0)		70 (24.1)		0.007
RIC	15 (4.9)		1 (5.55)		14 (6.7)		
Acute GVHD occurrence	186 (60.4)		9 (50.0)		177 (61)		
severe (grade III–IV)	48 (15.6)		4 (22.2)		44 (15.2)		
Chronic GVHD occurrence	69 (22.4)		4 (22.2)		65 (22.4)		
severe (grade III–IV)	28 (9.1)		0 (0)		28 (9.7)		
High dose steroid administration after HCT *	170 (55.2)	sd	11 (61,1)	sd	159 (54.8)	sd	
Mean duration of steroid (days) [min–max]	90.8 (7–365)	70	46 (30–214)	70.5	91.3 (7–365)	70.3	
Immunosuppressive treatment used in 2nd line Treatment after high steroid administration	63 (20.5)		4 (22.2)		59 (20.3)		
Relapse post-aHCT	36 (11.7)		2 (11.1)		34 (11.7)		
Engraftment failure **	16 (5.2)		2 (11.1)		14 (4.8)		
		sd		sd		sd	
Mean neutrophil recovery [min-max]	22.9 (10–79)	9.3	19.3 (12–31)	6.2	23.2 (10–79)	9.4	
Antifungal prophylaxis	50 (16.2)		2 (11.1)		48 (16.6)		
Primary prophylaxis	23 (7.5)		1 (5.5)		22 (7.6)		
Secondary prophylaxis	27 (8.8)		1 (5.5)		26 (9)		

Abreviations: HCT: hematopoietic stem cell transplantation; ALL: acute llymphoblastic leukemia; AML: acute myeloblastic leukemia; CR2: complete remission 2; PBSC: peripheric bone stem cell; TBI: total body irridiation; RIC: reduced intensity conditionning; GVH: graft versus host disease; CTC: corticotherapy; IFI: invasive fungal infection; * Receiving prednisolone with a minimal dosage of 1 mg/kg/day for more than a week; ** Engraftment failure was defined when there was less than 40% of donor cell engraftment (chimerism) 3, 6 and 12 months after aHCT.

### 3.2. IFIs: Incidence and Characteristics of IFI Patients

There was a total of 20 IFIs among 18 patients (5.8%): 16 (5.2%) until day 180 post-HCT with a median range of 57 days [min 2–max 304]. Thirteen IFIs were classified as proven, seven as probable. The cumulative incidence of proven and probable IFIs was 6.5% (Table 2). Proven and probable IFIs occurred homogeneously over the ten-year study. Ten IFIs were aspergillosis (50%), seven were candidosis (35%), two were mucormycosis (10%) and one was trichodermosis (5%). One patient developed both aspergillosis (probable) and candidosis (proven) infection, and another one experienced two probable aspergillosis infections, respectively, 3 and 6 months post-a-HCT. The lungs were the most common site of IFIs (55%). Four IFIs (20%) were only fungemia and four (20%) were disseminated, which was defined by having at least two organ localizations (Table 3).

Among the 18 patients with IFIs, 72.2% (13/18) had an underlying malignant pathology, of which 38.9% (7/13) had a CR status ≥ 2. The majority of grafts were realized with an unrelated donor (66%), and 66% (12/18) of the children were over 10 years old. The majority of patients with an IFI (11/18–61%) did not have a complete lymphocyte reconstitution after one year.

Half of IFI patients developed an acute GvHD and all received a long, high-dose steroid therapy for more than 10 days. Among them, 44% (4/9) had grade III–IV. The occurrence of chronic GvHD was 22% (4/18), but none of these cases were severe.

Among the 10 patients who developed late IFIs (beyond 2 months), 70% presented a GvHD: 40% of acute GvHD and 30% of chronic GvHD. Only one of the patients received an additional immunosuppressive treatment before the occurrence of his IFI.

Concerning viral infections, seven patients had a CMV reactivation or disease and five of them had a co-infection with EBV; there were no cases of EBV infection alone. Almost 10% of patients had an HHV6 viral reactivation or adenoviral infection (Table 1 and Table 3).

### 3.3. Determinants of IFI Identified by Logistic Regression Analysis (Table 4)

Univariate analysis found that an age over 10 years (OR: 0.29; *p* = 0.014) and the low neutrophil count (OR: 0.93; *p* = 0.01) were significantly associated with a risk of IFI occurrence. There was also a trend for malignant underlying disease (OR: 2.02; *p* = 0.173), a complete remission (CR) status ≥ 2 or more (OR: 2.25; *p* = 0.107) and the use of a therapeutic antiviral treatment (OR: 2.39; *p* = 0.07). To be noted, antifungal prophylaxis, GvHD and immunosuppressive drugs (including corticosteroid therapy) did not show any significant relationship with IFI in our study. There was also no risk factor found to be fungal organism-specific or period-specific.

**Table 4 cancers-15-02107-t004:** Possible risk factors for the occurrence of IFI after pediatric allogenic HCT after one year of monitoring.

Risk Factors	No. of Patients	Invasive Fungal Infection (IFI)			Univariate Analysis			Multivariate Analysis	
		N	%	OR	95% CI	*p*-Value	OR	95% CI	*p*-Value
Age at SCT									
>10 years old	120	12	10	0.29	0.10–0.81	0.014	0.29	0.09–0.83	0.022
Malignant underlying disease	176	13	7.4	2.02	0.70–5.83	0.173			
≥CR2 or more	83	7	8.4	2.25	0.85–5.91	0.107			
CMV risk reactivation *	42	0	0	NA	NA	NA			
CMV reactivation or disease	96	7	7.3	1.42	0.53–3.79	0.486			
ADV disease	20	2	10	1.87	0.39–8.79	0.456			
No neutrophils engraftment (counted as continous variable)	8	2	25	0.93	0.87–0.98	0.011	0.93	0.87–0.99	0.024
No lymphocyte reconstitution (counted as continous variable)	95	11	11.6	0.99	0.98–1	0.018			
Type of donor									
Matched related donor	106	6	5.7	0.9	0.3–2.2	0.78			
Unrelated donor	188	12	6.4	1.1	0.5–2.8	0.79			
Haplo-identical donor	14	0	0	NA	NA	NA			
Stem cell source									
Bone marrow	229	11	4.8	0.6	0.2–1.5	0.25			
PBSC	19	2	10.5	1.6	0.3–7.4	0.6			
Cord blood unit	56	5	8.9	1.7	0.6–4.7	0.29			
Mixed	4	0	0	NA	NA	NA			
Type of conditionning									
Myeloablative regimen	293	17	5.8	0.4	0.1–2.2	0.382			
Acute GVHD occurrence	186	9	4.8	0.79	0.31–1.96	0.612			
severe (grade III–IV)	48	4	8.3	1.39	0.44–4.34	0.585			
Chronic GVHD occurrence	69	4	5.8	1.4	0.5–3.6	0.54			
severe (grade III–IV)	28	0	0	NA	NA	NA			
Total immunosuppressive treatment used other than cyclosporine after HCT **	233	15	6.4	1.4	0.5–3.6	0.54			
High dose steroid administration after HCT	170	11	6.5	1.48	0.57–3.83	0.405			
Antifungal prophylaxis	50	2	4	0.79	0.25–2.45	0.681			
With prior history of IFI	27	1	3.7	0.53	0.06–4.11	0.507			
Total antiviral drugs	245	18	7.3	1.6	0.6–4.3	0.297			
Rituximab	101	6	5.9	1.2	0.5–2.9	0.72			
Curative Antiviral treatment ***	144	12	8.3	2.39	0.87–6.55	0.079	2.71	0.92–7.99	0.069

Abreviations: HCT: hematopoietic stem cell transplantation; ALL: acute llymphoblastic leukemia; AML: acute myeloblastic leukemia; RC2; PBSC: peripheric bone stem cell; TBI: total body irradiation; RIC: reduced intensity conditionning; GVH: graft versus host disease; CTC: corticotherapy; IFI: invasive Fungal infection; CMV: Cytomegalovirus; ADV: adénovirus; CR2: complete remission 2. * CMV risk reactivation was defined as mismatched CMV serology with donor + and recipient. ** Immunosuppressive treatment including high steroid after HCT (receiving prednisolone with a minimal dosage of 1 mg/kg/day for more than a week) and other immunosuppressive drugs used in second line after steroid administration. *** Curative antiviral treatment except Aciclovir.

In the multivariate analysis, the final model retained (best likelihood/lowest AIC) three parameters that predicted IFIs: aged over 10 years (OR: 0.29; [0.09–0.83]), slow and incomplete reconstitution of PNNs (OR: 0.93; [0.87–0.99]), and use of therapeutic antiviral drugs (OR: 2.71; [0.92–7.99]).

### 3.4. Secondary Outcomes

The one-year overall survival (OS) rate in our cohort was 86%. The total mortality rate among the IFIs population was 44% (8/18), but only 6 children died due to an IFI which represents an IFI mortality-rate of 2% (6/308) in our cohort and 14% (6/43) among all the other causes of deaths. The main other causes of deaths were, respectively, relapse for 27.9% (12/43) of patients, infectious diseases (except IFIs) for 20.9% (10/43), GvHD for 20.9% (9/43), veno-occlusive diseases (2/43), refractory auto-immune diseases (2/43) and other causes (2/43). Among the 18 allografted patients with an IFI, 10 had a stay in the intensive care unit (55%) and 2 relapsed (11%).

### 3.5. Antifungal Prophylaxis

In our population, only 16.2% received antifungal prophylaxis of whom 7.5% (23/308) received a primary antifungal prophylaxis. Among these 23 patients, 7 received a primary prophylaxis due to construction work very close to the stem cell transplant unit during 2012. For the others, it was essentially patients with lung malformations (pneumatoceles, bronchiectasis), some patients with severe aplastic anemia, Fanconi anemia or young adults (>17 years old) (Table 1).

Almost 8.8% (27/308) of patients received secondary antifungal prophylaxis due to prior IFI history before a-HCT. Only 37% (10/27) had refractory disease (CR status > 2) with a-HCT representing their only chance for therapeutic therapy. Twelve had a history of invasive pulmonary aspergillosis, thirteen had a history of invasive candidosis infection and two had a history of disseminated fusariosis infection. Nine (33%) had a stay in intensive care unit, four (15%) relapsed and seven died (25%). Only one patient presented another IFI post-a-HCT (trichodermosis infection).

In total, antifungal prophylaxis was administered in 50 HCT procedures (23 in primary and 27 in secondary prophylaxis). There was no clear tendency in prophylaxis choice: 56% of echinocandin (caspofungin (*n* = 21), micafungin (*n* = 7)); 40% of azole treatment (posaconazole (*n* = 12), voriconazole (*n* = 7), fluconazole (*n* = 1)); 4% of liposomal amphotericin B (*n* = 2). Fluconazole represented less than 5% of all antifungal prophylaxis. (Figure 1).

Among the eighteen patients with IFIs, only one patient received primary prophylaxis and another one received secondary prophylaxis. In the first case, the patient received posaconazole and finally presented trichodermosis 7 months post-a-HCT. Posaconazole spectrum does not cover this type of fungal organism. In the other case, the patient had a prior history of proven *Aspergillus flavus* infection 3 years before a-HCT, treated by 3 mg/kg/day of liposomal amphotericin B (L-AmB). He then received a prophylactic dose of 10 mg/kg/week of L-AmB. However, despite the action of L-AmB on molds, the patient developed two probable aspergillosis, 3 and 6 months post-a-HCT.

The cumulative incidence of IFIs revealed that whatever the presence or the type of prophylaxis, IFIs occurred mainly within the first 6 months post-a-HCT. The incidence decreased progressively. No significant difference has been shown between patients with or without primary prophylaxis (*p* = 0.462).

## 4. Discussion

This 10-year retrospective study presents the results of our center regarding the incidence of IFIs for children who underwent a-HCT without a systematic use of antifungal prophylaxis treatment. There is currently a major concern regarding the optimal strategy for the use and the choice of the antifungal drug. However, other significant factors must be considered, such as local ecology, appearance of fungal resistance, toxicity, drug interaction, specific pediatric pharmacokinetics and cost.

The environmental prophylaxis used during all the HCT procedures is fundamental in the inpatient setting with a very high risk of IFIs during all the neutropenia periods. However, when patients leave the protected unit, the environmental prophylaxis in the outpatient setting consisted only of parental therapeutic education to secure and control the environment at home. This point is essential during the post-HCT period when patients remain at risk of developing IFDs.

The rate of IFIs during the first year post-a-HCT in our study was consistent with the literature, generally reported between 5 and 25% in a similar pediatric population [4,5,6,7,8,9,10,11,12,13,14]. However, in their recent study of 290 pediatric patients with a median age similar to our cohort, Dvorak et al. reported an IFI incidence of 1.2% among patients who benefited from systematic primary prophylaxis with an azole (fluconazole or voriconazole) or echinocandins during the time until discharge from hospital for HCT, which represents only one of the periods of risk for developing IFI [15]. In our cohort, no difference was observed depending on the use or not of a primary prophylaxis. As a reminder, only 7.5% of our patients received primary antifungal prophylaxis and 8.8% received secondary prophylaxis. Our IFI-related mortality rate was lower than those reported in other studies (60% to 80%) [4,5,6,7,8,9,10,11,12,33,34]. This study reinforces the hypothesis that not systematizing antifungal prophylaxis in a-HCT pediatric patients could be performed.

As previously described, the most frequently encountered fungal organism was *Candida* spp. probably due to the difficulty to isolate *Aspergillus* spp. In the literature, we observed the same type of fungal organism distribution [14,25]. One case of trichodermosis and two cases of mucormycosis occurred (two of which were fatal) and two patients had a prior history of fusariosis before their a-HCT. The literature also identified this trend of increasing atypical cases [25], with the appearance of resistance to antifungals. Local ecology and the proper use of antifungals are very important, especially since some children need three lines of treatment to fight their IFIs.

In our study, IFIs occurred mainly during the first 6 months, regardless of the use of prophylaxis, and rather in accordance with the kinetics of hematopoietic recovery, suggesting that few IFIs would have occurred beyond one year. Multivariate analysis confirms that an age over 10 years is a significant risk factor for IFIs. This association of age over 10 years and occurrence of IFDs is important to note. This finding is in accordance with other similar observations made for ALL patients. The relation between cellular reconstitution and IFIs was significant but needs to be nuanced as the confidence interval was close to 1. In addition, as already described in the literature, the risk of IFI increases when the a-HCT occurs at CR status ≥ 2 versus after a first-line treatment [14,22].

Based on prior studies, it is now recognized that grade II-IV acute GvHD, extensive chronic GvHD, prior history of IFI, high-dose corticosteroid administration post-HCT for more than a week, severe aplastic anemia or Fanconi anemia, relapse and secondary neutropenia are frequent risk factors for IFIs in children undergoing a-HCT [9,35]. However, our study did not show significant risk of IFI occurrence with GvHD and immunosuppressive treatment (high steroid therapy or other drugs except ciclosporin). This is probably a limit of retrospective study without available time-to-event detailed data. These results must be verified by another strong statistical model using time-to-event modeling. In their paper, Lehrnbecher et al. suggested that antifungal prophylaxis administration in patients receiving systemic treatment of GVHD could be reasonable based on risk factors for IFD [13]. However, in our study, only 5% of patients suffering from acute GVHD and chronic GVHD developed an IFI.

Moreover, to our knowledge, it is the first time that a study suggests that therapeutic antiviral treatment is a significant risk factor for IFIs. One hypothesis could be that depletion of the immune system is due to viral infections and their treatments, in already immunosuppressed patients. Induced neutropenia and lymphopenia are two well-known side effects of antiviral drugs such as valgangiclovir, ganciclovir or rituximab. Antivirals could have prolonged neutropenia recovery or lymphocyte reconstitution and predisposed patients to IFIs. The majority of patients (79.5%–245/308) received at least a therapeutic antiviral treatment and this was the case for all IFI patients. As a reminder, most patients with an IFI (61%) did not have a complete reconstitution of lymphocytes after one year of monitoring versus 49% of patients with no IFI. Moreover, among patients with no IFI, 47 patients had a viral infection or a viral reactivation without receiving any therapeutic antiviral treatment. Moreover, viral infections often reactivate during increased immunosuppression, so their impact on the occurrence of IFIs remains unclear. This could suggest that there was probably a double implication of viral infection and therapeutic antiviral treatments. It would be interesting to confirm this result with other prospective studies.

Therapeutic management procedures have remained unchanged in the last 10 years. In our cohort, there was a trend for choosing echinocandin when a primary antifungal prophylaxis was introduced, but it remained at the discretion of the respective attending physician. The purpose of this study work was also to conduct an overview of our practices and harmonize them. There was also no uniform practice on the national territory [25], but no additional risk either of IFIs observed with one antifungal or another [14]. A systematic review by Lehrnbecher et al. [13] included 68 randomized trials and made strong recommendations to administrate systematic antifungal prophylaxis to children and adolescents receiving treatment for acute myeloid leukemia, as well as to those undergoing a-HCT pre-engraftment, and those receiving systemic immunosuppression for graft-versus-host disease treatment. However, this had moderate-quality evidence as only six (9%) trials were conducted in a solely pediatric population. The authors recognize that the adult data were clearer, and that these data may be less generalizable to pediatric patients because of different transplantation approaches such as stem-cell source. It should also be considered that young children are usually less colonized than adults, and therefore less likely to develop IFI.

Fluconazole is not active on molds. If an introduction of prophylaxis is necessary, the choice of molecule should be considered case-by-case: echinocandins cause fewer adverse effects and interactions than azoles, but azoles such as voriconazole or itraconazole can be taken orally. Since 2022, new European guidelines have just also authorized the use of the posaconazole oral suspension form in children older than 2 years for antifungal prophylaxis. With an azole, therapeutic drug monitoring can be realized, allowing us to adjust the doses. Furthermore, the use of azoles, of which posaconazole is one example, could be more interesting because of their spectrum on *Aspergillus* spp. and other atypical emerging fungal organisms.

## 5. Conclusions

Our study suggests that systematic antifungal prophylaxis for children undergoing a-HCT should be discussed to prevent IFI occurrence according to ECIL-8 recommendations [22]. An environmental prophylaxis, such as the one established in our tertiary reference center, seems to be an efficient measure to reduce IFI rate. A discussion could be initiated about ensuring systematic primary antifungal prophylaxis only for children with high IFI risk factors, such as those over 10 years of age associated with a remission status ≥ 2, but not systematically, as is the case for the adult population. However, the use of any antifungal drug must be informed by local ecology. It is also important to mention that parental therapeutic education plays a leading role in the care pathway for allografted children especially when they leave the protected unit to secure and control the environment at home. The best results in the prophylaxis of IFI are based on the association of these integrative measures. To our knowledge, this is the first large-scale French study conducted only on a pediatric cohort that highlights these results.

## Figures and Tables

**Figure 1 cancers-15-02107-f001:**
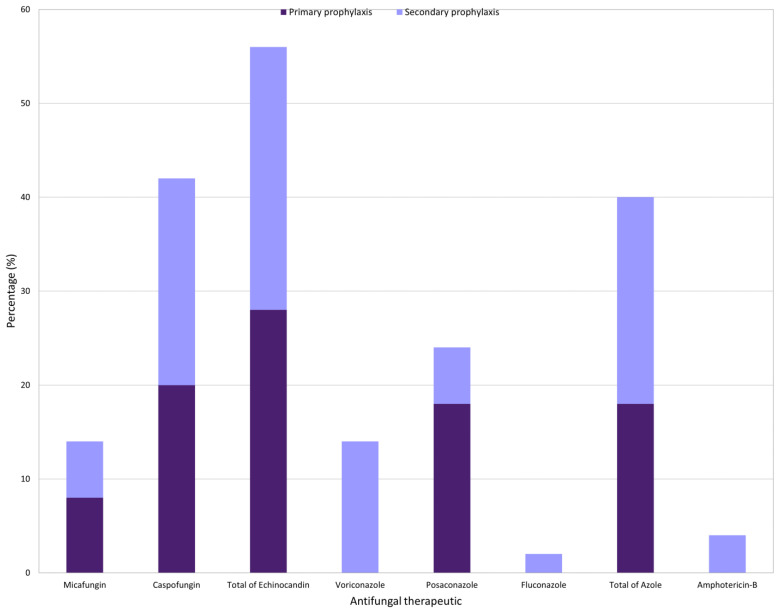
Overview of antifungal prophylaxis in 50 HCT procedure among 308 a-HCT patients.

**Table 2 cancers-15-02107-t002:** Incidence of early and late IFIs depending on to the fungus among the 308 pediatric aHCT patients.

Fungi	Early IFIs		Late IFIs		Total	
	n	%	n	%	N	%
*Candidosis*	7	2.3	0	0	7	2.3
Aspergillosis	7	2.3	3	1	10	3.3
*Mucormycosis*	2	0.6	0	0	2	0.6
*Trichodermosis*	0	0	1	0.3	1	0.3
Total	16	5.2	4	1.3	20 *	6.5

Early IFI if occurs from day 0 to day 180 after a-HSCT; Late IFI if occurs from day 181 to day 365 after a-HSCT; * Total incidence of IFI 18/308 (5.8%) but one patient developped one IFI with candida and aspergillus combined (+1) and one patient developped two Probable IFIs (+1).

**Table 3 cancers-15-02107-t003:** Clinical charecterisitcs and outcome of patients with probable and proven IFIs.

N°	M/F	Age (Years)	Underlying Disease	Status before HSCT	History of IFI Prior HSCT	IFI Classification	Fungus	HSCT—IFI Interval (Days)	Location	Treatment	Antifungal Prophylaxis	Intensive Care	Outcome After One Year of Monitoring
1	M	13.4	B-ALL	CR3	No	Proven	Mucormycosis (rhizomucor)	J20	Lung	Amphotericin-B, Posaconazole	No	Yes	Deceased
2	F	15	SAA/HPN	_	No	Proven	*Aspergillus fumigatus*	J62	Disseminated	Caspofungin	No	Yes	Deceased
3	F	13.4	Hodgkin	CR2	No	Proven	*Candida dubliniensis* *Candida albicans*	J29	Disseminated	Amphotericin-B, 5-FU	No	Yes	Deceased
4	F	0.8	SCID	_	No	Proven	*Candida albicans*	J166	Blood	Caspofungin	No	No	Alive
5	F	16.2	Lymphoblastic T Lymphoma	CR2	No	Proven	*Aspergillus flavus*	J37	Lung	Caspofungin	No	Yes	Deceased
6	F	7.4	Fanconi Anemia	_	No	Proven	*Candida parapsilosis*	J66	Blood	Fluconazole	No	Yes	Deceased
7	F	15	AML	CR1	No	Proven	*Candida dubliensis*	J5	Blood	Caspofungin, Fluconazole	No	No	Alive
8	M	3	SAA/MDS	_	No	Proven	*Mucormycosis*	J133	Disseminated	none (palliative cares)	No	Yes	Deceased
9	M	18	Acute CML	CR1	No	Proven	*Trichodermosis*	J213	Lung	Voriconazole	Primary (Posaconazole)	Yes	Alive
10	M	14	B-ALL	CR1	No	Proven	*Candida tropicalis*	J48	Lung	Fluconazole	No	No	Alive
11	F	10	Anaplasic Lymphoma	CR1	No	Proven	*Candida Kefyr*	J6	Blood	Amphotericin-B, Caspofungin	No	No	Alive
12	M	16.9	Hodgkin	CR2	No	Proven	*Candida glabrata* *Candida albicans*	J34	Lung	Caspofungin	No	No	Alive
13	F	8	MDS		No	Proven	*Aspergillus fumigatus*	J304	Intestinal	Amphotericin-B	No	Yes	Deceased
14	F	13.4	Hodgkin	CR2	No	Probable	Aspergillosis	J29	Disseminated	Amphotericin-B, 5-FU	No	Yes	Deceased
15	F	19.75	T-ALL	CR2	Yes	Probable	Aspergillosis	J90	Lung	Voriconazole	Secondary (Amphotericin-B)	Yes	Alive
16	F	19.75	T-ALL	CR2	Yes	Probable	Aspergillosis	J183	Lung	Amphotericin-B	Secondary (Amphotericin-B)	Yes	Alive
17	M	2	JMML	CR2	No	Probable	Aspergillosis	J2	Lung	Amphotericin-B	No	No	Deceased
18	M	16.3	SAA	_	No	Probable	Aspergillosis	J152	Lung	Voriconazole	No	Yes	Alive
19	F	12	AML	CR2	No	Probable	Aspergillosis	J52	Lung	Voriconazole	No	No	Alive
20	F	12.9	B-ALL	CR1	No	Probable	Aspergillosis	J244	Lung	Voriconazole, caspofungin	No	No	Alive

Abbreviations: M: male; F: female; B-ALL: B acute lymphoblastic leukemia; SAA: severe aplastic anemia; HPN: paroxysmal nocturnal hemoglobinuria; SCID: severe combined immunodeficiency; AML: acute myeloblastic leukemia; MSD: myelodisplastic syndrome; CML: Chronic myeloid leukemia; JMML: juvenile myelomonocytic leukemia; CR: complete remission; IFI: invasive fungal infection; HCST: hematopoietic stem cell transplantation.

## Data Availability

Informed consent has been obtained from the legal representatives of the children before all stem cell transplantation procedure and we included only patients who agreed to participate in retrospective studies, in accordance with the MR004 law.

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
