# Peer review of "New Perspectives on Primary Prophylaxis of Invasive Fungal Infection in Children Undergoing Hematopoietic Stem Cell Transplantation: A 10-Year Retrospective Cohort Study"

_cancers, 2023, doi:10.3390/cancers15072107_

Round 1

Reviewer 1 Report

The manuscript submitted by Ricard and colleagues reports the results of a retrospective single center analysis of a strategy of no systematic antifungal prophylaxis in children undergoing allogeneic HCT with an observation period of one year after transplant. Among 308 patients, eighteen developed twenty IFDs (13 proven, 7 probable; 6.5%), and 2% of the children died with causal relationship to an IFD. The authors conclude that IFD occurrence and mortality from IFDs was not higher than what is reported in literature with the use of systematic antifungal prophylaxis, and suggest focusing prophylaxis on high risk patients that were those over 10 years of age in their multivariate analysis of risk factors.

I have the following comments and/or suggestions:

1.         English syntax needs to be checked throughout and editing of the manuscript by an individual proficient in medical English is recommended.

2.         Pl. replace the term HSCT by HCT, hematopoietic cell transplantation, which is the term favored by the EBMT.

3.         Introduction: Pl. note that ECIl-8 recommends several options for prophylaxis, and not only posaconazole in the post engraftment phase.

4.         Introduction: Antifungal prophylaxis according to ECIL-8 is indicated in patient populations with a natural incidence rate of 10% and higher and this generally includes allogeneic HCT recipients. However, as stated in the guideline, the local epidemiology is an important consideration to design local approaches to management of IFDs, and considering patients individual risk factors also is essential for best management.

5.         Introduction: Please note that while environmental prophylaxis may be useful in the inpatient setting, it is not applicable in the outpatient setting, where the risk to develop IFDs continues.

6.         Methods: It is interesting to note that prophylactic use of antibiotics in your program is quite intense, and the reviewer is wondering how this fits into the stewardship concept of avoiding antifungal agents. What is your overall concept regarding antimicrobial stewardship? Pl. clarify.

7.         Methods: Please provide your SOPs regarding empirical and pre-emptive antifungal therapy, antifungal agents used for these indications, and the routine algorithms for making the diagnosis of probable or proven IFDs. The incidence of probable and proven IFDs in a given institution depends on the efforts to make this diagnosis with appropriate imaging and microbiological investigations.

8.         Methods: Pl. replace the term curative by the term therapeutic (as opposed to prophylactic).

9.         Methods: Why did you stop to assess outcome at 1 year post transplant and did not also include long term survival to the last follow-up?

10.     Methods/results, line 153 and table 2: The usual definition of phases at risk is from the start of the conditioning regimen until engraftment; and post engraftment until day +180 and after day +180. Please adhere to this conventional classification.

11.     Methods/results:  Apart from prophylaxis, utilization of antifungal agents in the post-transplant setting for empirical and pre-emptive therapy is considerable - we may refer the authors to the Linke et al. paper in MYCOSES  that contains an example of an analysis of antifungal consumption in pediatric allogeneic HCT recipients in a real life setting. For assessment of your strategy, it would be helpful to know the complete picture of antifungal usage, i.e. the % of patients who received either empirical, preemptive, or targeted antifungal therapy, and the overall % of days in hospital in which patients are on antifungal therapy.

12.     Results: The association of age over 10 years and occurrence of IFDs is interesting and important as similar observations have been made for ALL patients.

13.     Results: Please provide information on the agents administered in the 16% of patients receiving antifungal prophylaxis and your rationale – I see caspofungin, which, however, is not approved in this indication and for which existed no data for most of the time of your survey. Pl. clarify your decision making.

14.     Discussion: Pl. note that prophylaxis in the Dvorak paper included only the time until discharge from hospital, and thereby only part of the period at risk.

15.     Discussion: Please replace the term germ by the term fungal organism.

16.     Discussion: Pl. note that viral infections often reactivate during increased immunosuppression, so that an independent impact on the occurrence of IFDs remains unclear.

17.     Discussion: Pl. consider the current approval status of posaconazole in the E.U. – the compound is approved in children 2 years and older for prophylaxis.

Reviewer 2 Report

Dear authors,

Your paper is interesting, timely and significant. Manuscript covering the issues for which there are not sufficient data in the literature. Guidelines prophylaxis and treatment of IFI in pediatric patients has to be harmonized and improved. Frankly, you’re presented the study for which it is necessary to invest a lot of effort and I have read this paper with interest

However, you have to improve the paper for publication.

My general suggestion is to include medical mycologist, since your listing of either the names of the fungi or the fungal infection they cause is incorrect. For example:

- your statement (Abstract): among which Aspergillus 30 spp (n=10, 50%) and Candida spp (n=7, 35%) were the most frequently identified organisms is wrong because in your study you did not identified all fugal causative agents, you had diagnosed proven or probably candidosis or aspergillosis

- Mucormycosis has to be Mucorales group of fungi

- Fusarium sp. has to be written, in some part of text it is furariosis (page 8, line 214)

-You must separate the species names from the terms of the diseases they cause.

Generally,it is very important to be written the methodology used for diagnosis of IFI, briefly of course. What was direct examination of proven IFI? Related to the topic it is important to discuss the application of amphotericin B in procedure of digestive decomamination since that the digestive tract is the main source of infection in candidemia and invasive candidosis.

My opinion is that it should always be emphasized that integrative measures give the best results in prophylaxis of IFI.

Round 2

Reviewer 1 Report

I have no further comments or recommendations.

Reviewer 2 Report

Dear authors,

For mycological point of view, you have to make some specific changes

In your study you did not change text in summary where I suggested  that species can be identified but disease can be diagnosed-

So, the sentence among which aspergillosis 29 (n=10, 50%) and candidosis (n=7, 35%) were the most frequently identified organisms must be change to

among which aspergillosis 29 (n=10, 50%) and candidosis (n=7, 35%) were the most frequently diagnosed infection.

Moreover, in page 4, line 175 where you mention fungal infection you have to change Mucorales fungi  to mucormycosis, and in page 5, line 228 you still write infection caused by Fusarium spp. as furariosis, it is fusariosis.
